# Exploring the Perspectives of South African Parents and Primary Caregivers Living in Low-Income Communities on What Children Need to Thrive within the First 1000 Days of Life

**DOI:** 10.3390/children8060483

**Published:** 2021-06-07

**Authors:** Babatope O. Adebiyi, Tessa Goldschmidt, Fatiema Benjamin, Inge K. Sonn, Nicolette V. Roman

**Affiliations:** Centre for Interdisciplinary Studies of Children, Families and Society, University of the Western Cape, Bellville 7535, South Africa; 3302887@myuwc.ac.za (T.G.); fbenjamin@uwc.ac.za (F.B.); isonn@uwc.ac.za (I.K.S.); nroman@uwc.ac.za (N.V.R.)

**Keywords:** parenting, childrenneeds, first 1000 days, children, low-income communities, South Africa, parents, primary caregivers, mother, father

## Abstract

The first 1000 days is recognised as a critical period for the development of children. What children need to thrive in this particular phase of development may be different from any other phase. In South Africa, parents’ perception of children’s needs within the first 1000 days of life could be considered as emerging. Therefore, this study aims to explore the perspectives of South African parents and primary caregivers on what children need to thrive within the first 1000 days. An exploratory qualitative study design was used to explore the parents’ understanding of what children need to thrive in the first 1000 days. A purposive sampling approach was employed to select parents and primary caregivers in low-income communities. In all, thirty respondents participated in the study. The data were analysed using thematic analysis. During the analysis, four themes emerged. The themes included (1) the importance of parenting, care and support; (2) children’s need for holistic development; (3) parental roles; and (4) sharing responsibilities. Parents and primary caregivers living in low-income communities understand what children need to thrive within the first thousand days of life. The study could assist policymakers and service providers to design appropriate interventions for parents within these communities.

## 1. Introduction

The Lancet Series of 2008 acknowledged the need to focus on the crucial period of pregnancy and the first 2 years of life–first 1000 days of life [1,2,3,4,5]. The first 1000 days can be described as the period from conception through to the age of 2. It is also a critical period for the development of childhood diseases [6]. The importance of the first 1000 days initiative has been recognised around the world to meet the needs of children early in life. In South Africa, the importance of the first 1000 days initiative is becoming increasingly recognised [7,8]. The National Integrated Early Childhood Development Policy of South Africa supports the first 1000 days initiative by providing policy backing for the initiative [9]. 

South African provinces such as the Western Cape have started first 1000 days initiatives which include a holistic programme promoting the well-being and meeting the needs of mothers and their children in communities. These initiatives aim to address the five major biological risk factors for loss of developmental potential that have been identified among children in the first 1000 days [10]. The five major biological risk factors include malnutrition (particularly stunting and anaemia), Human Immunodeficiency Virus (HIV) infection and exposure, alcohol exposure, methamphetamine exposure and traumatic brain injury [10]. These factors may contribute to school drop-out and unemployment later in life [10]. Furthermore, studies have reported that over 250 million children under the age of five years, living in low- and middle-income countries may not reach their developmental potential due to the risk factors mentioned above [11,12]. Therefore, parents need to understand what children need to thrive. 

Addressing the risk factors for loss of development potential among children within the first 1000 days requires a holistic approach. The Lancet 2008 Series has called for better national priority for nutrition programmes, improved intersectoral approaches and greater coordination in the global nutrition system of international agencies, donors, academia, civil society and the private sector. International organisations such as United Nations Children’s Emergency Fund (UNICEF) and World Health Organization (WHO), have recognised parenting practices to have a positive impact on the child’s overall developmental outcomes, particularly in the first 1000 days [13]. Moreover, these organisations have identified parenting education programmes, in particular, as a priority for improving children’s development outcomes in low- and middle-income countries [13]. These educational programmes may enable parents to understand what children need to thrive within the first 1000 days of life. 

The Nurturing Care Framework provides a summary of what children need to thrive and transform, which includes good health, nutrition, security and safety, responsive caregiving and opportunities for early learning. According to the Nurturing Care Framework, adequate nutrition formed an important aspect of children’s needs in the first 1000 days of life [14,15]. The effect of malnutrition is severe during the first 1000 days because growth and development occur more rapidly during pregnancy and infancy than in any other period over the life span [16]. Besides, maternal and infant nutrition within the first 1000 days has been linked with holistic development, infants’ later physical growth, muscle mass development, brain development, cognitive functioning, socio-emotional adjustment, risk-taking behaviour, earning capacity and a multitude of health-related concerns [16,17,18,19,20,21,22]. 

Cognitive stimulation is another need of children [14,15]. Opportunities for early learning such as reading to the child during the first 1000 days has been associated with improved language development, mathematical and problem-solving capacity and higher earning capacity later in life [23,24,25]. In addition, studies have shown that parental warmth, nurturance and responsiveness during the first 1000 days of life, affect infants’ emotional regulation, capacity for wealth creation, behavioural problems, health and cognitive development throughout their lifespan [26,27,28,29]. Therefore, the nurturing care and support infants receive within the first 1000 days of their lives will either enable or limit their capacity as they grow into adolescents and adults. 

Additionally, children need responsive parenting–an aspect of nurturing parenting–which has been identified to play a vital role in providing a strong foundation for children’s optimal development [30,31]. There is a strong link between responsive and nurturing parenting and an increased volume in the brain regions responsible for the regulation of stress in normal developing children [32]. Furthermore, a randomised control study found that if children do not receive nurturing care from a safe home environment, they develop poor long-term childhood outcomes such as conduct disorders and gang violence [33]. Moreover, security and safety also form part of the children’s needs [14].

It is essential to identify what children need to thrive in the first 1000 days so that they can fulfill their human potential. Children are dependent on parents to reach their potential which means that parents need to know how to ensure that their children thrive. A study conducted in the United State of America identified the needs of children and suggested the need to educate parents on the care of young children [34]. The need for education could mean that parents may not understand what children need within the first 1000 days, and this may be more of a need in low-income communities. Another study in Europe has reported that children need parents as educators [35]. In addition, the educational and psychological needs of children in Asia were reported in a study [36]. In South Africa, a study reported what children need within the first 1000 days, but the study explored the perspectives of community health workers [15] and not parents, and not parents in low-income communities. This is therefore the first known study, specifically focusing on the perspectives of parents and primary caregivers in low-income communities, on what children need to thrive within the first 1000 days in South Africa. The focus on low-income communities is another unique contribution of this study as often parents in these communities are more vulnerable than parents in resource filled communities. The main research question is, what do children need to thrive within the first 1000 days of life?

## 2. Materials and Methods

### 2.1. Study Setting

This study was conducted in low-income communities of the Western Cape Province of South Africa. These low-income communities included Khayelitsha, Saldanha, Caledon, Mitchells Plain, Manenberg, Grabouw, Fisantekraal, Vredenburg, Genadendal and Lamberts Bay. Khayelitsha is a township located in Cape Town, South Africa. Khayelitsha is one of the townships that were established under apartheid law to house black African populations. Khayelitsha comprises a population of almost 400,000 living in approximately 120,000 households with an average of 3.3 people per household according to the 2011 census data [37]. Khayelitsha is an exciting, densely populated and unified community beset by high levels of poverty, unemployment, and poor service provision [38]. Around a third of adults have matriculated (36%) with three quarters of households having monthly incomes of R3 200 or less [37]. Khayelitsha and other townships mentioned above were selected because they have similar characteristics (low-income households) and met our inclusion criteria. 

### 2.2. Study Design

The exploratory qualitative research design was used to explore the parents’ and caregivers’ understanding of what children need to thrive in the first 1000 days. This research design was chosen as the parents’ perception of children’s needs within the first 1000 days could be considered as emerging in South Africa. The exploratory qualitative research design was used to gain a deeper understanding of what children need to thrive within the first 1000 days [39].

### 2.3. Sampling Procedure

A purposive sampling approach was employed to select the target participants. In applying this approach, a door-to-door process was employed. The participants were selected if they were parents or primary caregivers of children younger than the age of two years. In addition, participants were included if they could speak and understand English, Afrikaans or isiXhosa and lived in low-income communities of the Western Cape. These areas were selected and the recruiter (the research team and research assistants) approached the potential participants, and participants were selected if they met the above criteria and if they were willing to participate. Table 1 provides information about the participants. In all, thirty participants participated in the study. The majority of participants were female, between the ages of 16 and 30 years, with an educational level of secondary school. Data saturation was reached when no ‘new’ information was elicited while conducting the interview [40].

### 2.4. Ethical Considerations

Approval was obtained from the Humanities and Social Sciences Research Ethics Committee at the University of the Western Cape (HS17/6/15). Participants were informed that participation was voluntary and that they could withdraw at any time if they wished to do so. An information sheet, detailing the aim of the study and the role of participants, was given to participants in their home language (English, Afrikaans, or isiXhosa) before the start of the interview. Individuals who agreed to participate after receiving the information sheet were requested to sign a consent form. The research team ensured the confidentiality of all information obtained during the study by encrypting documents with a password on a computer. In addition, no information associated with any of the participants was reported in this manuscript.

### 2.5. Data Collection

After ethical approval and consent had been obtained, semi-structured interviews were conducted by the research team. The interviews were conducted in English, Afrikaans and isiXhosa between late 2018 and 2019. These were the languages of the participants. The interviews lasted between 30–60 minutes and were guided by an interview schedule. The participants were asked various questions related to the child’s need and the child’s health and well-being. Open-ended questions were used to start the interviews and follow-up questions to probe for additional information when necessary. All the interviews were audio-recorded with permission from the participants.

### 2.6. Data Analysis 

The data were analysed using thematic analysis [41]. All the interviews conducted were transcribed and translated into English when necessary. After transcription and translation, the researchers read and reread the transcript for familiarisation and to generate initial codes. The initial codes were reorganised to obtain refined codes. Codes with similar ideas were grouped to form subthemes, and those subthemes which had similar ideas, were further grouped to form the final themes. After obtaining the final themes, themes were defined and supported by quotations from transcripts.

### 2.7. Trustworthiness and Rigour of the Study

The rigour and trustworthiness were established through credibility, transferability, dependability, conformability and through a reflexive approach to the inquiry and analysis [42].

In this study, to ensure transferability, a detailed methodology (the study’s site, participants, and procedures used to collect data) was provided. Dependability was ensured by providing and describing in detail the exact methods of data collection, analysis and interpretation. In addition, the interview schedule was developed in consultation with the research team and based on the literature. Two members of the research team independently coded the transcripts and also met afterwards to discuss the findings, and agreements were reached by consensus. Moreover, the same interview schedule was used as a guide for all the interviews. Credibility was ensured through member-checking conducted at the end of each interview—a recap of the salient points that emanated from the interviews. A reflective journal formed part of the audit trail that was kept for the study. A reflective journal is a document which contains the discussions, deliberations and decisions made by the researchers when conducting research. In addition to an audit trail, verbatim transcripts of the participants’ responses were provided to ensure confirmability. In reporting this study, all the relevant aspects of the criteria for reporting qualitative research (COREQ) outlined by Tong, Sainsbury and Craig [43], were followed.

## 3. Results

The themes and subthemes obtained during the data analysis are presented below. The word parent will refer to both parents and primary caregivers. 

### 3.1. Theme 1: The Importance of Parenting, Care and Support

Children need care and support as reported by the participants. The participants expressed both the short- and long-term effects of care and support for a child within the first 1000 days of life. The parents understood what positive parenting in the first 1000 days could mean for child development and well-being now and later in life. They seemed to be in support of authoritative parenting–a type of parenting practice where the parent is nurturing and supportive, while still giving the child room to explore. They alluded to the need for parents to support their children for holistic development.

Good upbringing

Participants noted the importance of good upbringing for a child, as the lack thereof could result in long-term effects. Suggesting that a good upbringing consisting of sufficient care for the child could substantially contribute to the child’s life trajectory. This is presented in the quote below: 

*“Uhm, it’s very important to know, like having care for your child, because if she gets older and she didn’t get that caring from a mother then she can go off the rails and so”* (Female, 20 years old, Grade 12).

Child development

This subtheme emerged because of parents reporting on what it meant to them to support their children. Many alluded to being present in various components of child development. Suggesting that support, according to participants, occurs across different developmental spheres:

*“Support to me is being there, it can mean physically, mentally, emotionally, financially (yes)”* (Female, 23 years old, Grade 12).

Freedom of expression

Children need freedom to thrive. The below extract speaks to authoritative parenting, whereby the parent allows the child independence while being present to support the child when needed:

*“Care is just caring about the person they are, who they becoming, caring about how they feel and having the right to feel the way they feel and having the right to do what they do, just being there for them. Giving them the freedom to be themselves, make mistakes and all, rights and wrongs and all. Just being there to say okay I’m here if you need my help”* (Female, 23 years old, Grade 12).

### 3.2. Theme 2: Children’s Need for Holistic Development

This theme considers the needs of children for holistic development within the first 1000 days from parents’ perspectives. These needs are essential for the health and well-being of children. These needs include love, attention and being present in a child’s life, and the health of the parent–especially mental health. This is because mental health can negatively interfere with parenting responsibilities.

Parental love

Parental love, especially from both parents, was considered as a protective factor against substance abuse in the future.

*“She must receive motherly and fatherly love... That is important because in the future, she didn’t get that motherly love then she can for example end up doing drugs and misusing alcohol and uhm, go off the rails”* (Female, 20 years old, Grade 12).

Parental attention and involvement

For children to attain optimal development, they need attention and parents need to be involved. The lack of parents’ attention and involvement could be detrimental to children’s development. This is because they may seek attention from the wrong persons and places. The quotations below illustrate this:

*“A child needs attention, a child needs love. So that is what we do, because we both work during the week, we have a little time from the child and weekends we are together, everyone, then we give them attention and what they need”* (Male, 32 years old, Grade 9).

*“There’s a lot of parents who push their children aside and say ‘no man go there I don’t feel for you right now, I don’t have time for you right now, I am tired, I am this and I am that…’ and if you do not give attention to a child then a child goes and looks for it someplace else, go and look for love other places and you are then the cause that your child will one day be what you never wanted your child to become”* (Female, 63 years old, None).

Parental availability

The participants believed that parents need to be available for their children. They noted that this will foster stability, safety and healthy behaviour. They emphasised the importance of the availability of the two parents over one.

*“If both parents are there for the kid. They’ll have more uh, uh, stability that the parent is there for them. So, they grow up in that way, my mother is here, I’m safe, I can speak to her about anything”* (Female, 48 years old, Grade 10).

*“To be more, uhm, available as a parent to the child and to be, uhm, ja take note of the child’s behaviour and pick up if there’s any discrepancies, or anything wrong, and to be attentive”* (Female, 25 years old, Degree).

Healthy parents

Children need healthy parents for them to thrive and transform. The participants reflected on the importance of parents’ mental health and well-being in the holistic development of a child. This is evidenced in the quotes below:

*“If I was psychologically that, uhm, damaged person and I feel like some of us partially is because of our life experiences and stuff. But if we let that, if I let that get to me I don’t feel like I’d be able to fulfil my role as a mother like I should be”* (Female, 25 years old, Degree).

*“If I’m mentally not in a good space or place then affects how I am with them”* (Female, 23 years old, Grade 12).

Nurturing the child’s emotional development

Parents expressed that a child’s emotions need to be nurtured, which will facilitate the understanding and expression of emotion in the future. The child will know when to laugh and when to cry. They said:

*“I need to nurture that feelings of her, I don’t know how to interpret this now, I don’t know how to say this. I need to provide her with love, so that she will be able to love one day, and not be a robot. She needs to understand that it’s okay to be angry, it’s okay to be sad, it’s okay to be happy, it’s okay to love. And it obviously needs to match the scenarios, so I feel like it’s important for her to express her emotions…”* (Female, 25 years old, Degree).

*“I make an angry face if she doesn’t respond to me, then I tell her I am angry that she doesn’t respond to me or so… That makes her feel very good and she laughs and she tells me not to be angry”* (Female, 22 years old, Grade 12).

### 3.3. Theme 3: Parental Roles

This theme encapsulates, that children need parents to guide children’s moral or ethical behaviour, teaching and nurturing their child’s development. In addition, parents should feed, play, love, provide and care for their children (including hygiene). This theme was divided into two subthemes; parents as the teacher and parents as the provider.

#### 3.3.1. Subtheme 1: Parent as the teacher

Children need parents as teachers to thrive. The first teacher in a child’s life is the parent. Parents teach children about morals, (appropriate) behaviour, manners, and emotional regulation skills, which are often derived from the parents’ own experiences and upbringing.

Guiding a child’s moral/ethical behaviour

Parents provide guidance for their children on moral and ethical behaviour.

*“I teach her the truth.... I teach her what is right and what is wrong. I show her the things that are wrong and what is right”* (Female, 24 years old, Grade 10).

Morals and values 

The participants said parents are custodians of values and they have the ability to transfer these values from one generation to another. Children need to learn values from their parents. 

*“I think religiously, to guide her and I think with regards to my morals, to install the same morals that were installed in me by my parents, to install it in her and make sure it’s followed through. Knowing the difference between right and wrong, and enforcing it. And also, with that, this is not contradictory but I want to make her a liberal thinker uhm, educator, she must be educated”* (Female, 25 years old, Degree).

*“I discipline her a lot. Like she must have very good manners because I have very good manners, my mother taught me, my parents taught me very good manners and so forth. I would like to teach her very good manners, I’m already teaching her to have very good manners, and to respect older people”* (Female, 20 years old, Grade 12).

*“Like my mother, they raised me like this. I’m going to raise my children like this. It’s about how the trend goes”* (Female, 38 years old, Grade 12).

Teaching

Participants reported that parents have the ability to teach their children what is right or wrong. 

*“I must love the child and everything she does wrong. I must teach her but this is wrong and this is the right word and the right thing and so”* (Female, 19 years old, Grade 9).

#### 3.3.2. Subtheme 2: Parent as the provider

Playing and loving

Children need love from their parents. They also need their parents to play with them. These will enable them to thrive.

*“Playing with her, uhm, love her, and talk with her and so on”* (Female, 20 years old, Grade 12).

Feeding

Children need to be fed to enable the development of their children.

*“I fed him, bathed him everything, when he was sick, he took him to a doctor in the clinic”* (Female, 29, Grade 11).

Provision

Participants reported that children need to provide for by their parents.

*“It is to make sure that he has food, he bathes, cleans and gives him love”* (Female, 30 years old, Grade 11).

*“Thinking…...Buying food for my child, keeping the child clean and buying clothes for my child”* (Female, 42 years old, Grade 9).

Role model

A participant reported that children need parents, especially fathers to act as role models. 

*“He is a very good role model for her like a father role model can be. And he does everything he can to give her the best”* (Female, 20 years old, Grade 12).

### 3.4. Theme 4: Sharing Responsibilities

Participants enumerated how children need both parents. Parents need to share responsibilities for optimal child development irrespective of their living arrangement.

#### Co-parenting

Children need to be supported by both parents. This is evidenced in the extracts below.

*“The oldest’, no, no, I wouldn’t say he isn’t a good parent, a good father, not a bad father, he just isn’t [there]. you can not only come when there are problems. But the baby’s father, oooh he is very nice. He is fond of his little girl. We don’t see each other in the week, just weekends. But he lets me rest. He helps a person very nicely”* (Female, 38 years old, Grade 12).

*If he (father) sees I’m tired then he takes over at night, in the late hours of the night when the child wakes up for the first time then it’s me who is busy with the child and if the child wakes up the second time at night and then he keeps eye on the child, then he cries and he gives him bottles”* (Female, 63 years old, Highest education level not disclosed).

#### Co-provider

Participants said that fathers also provide for their children to meet their needs. They said:

*“He’s very helpful... He brings his share. I still get a lot of support from him”* (Female, 23 years old, Diploma).

*“He’s a very good father, he loves him very much. He’s a good provider yes”* (Female, 23 years old, Grade 12).

#### Bonding

Bonding with parents is a need of a child. Both parents need to find time to bond with the child. This will enhance the parent-child relationship. This is obvious in the quotations below.

*“If I leave her at school then she cries, but if he leaves her at school then she’s fine. In the morning, when they walk to school everything is fine. She says pa pa pa. She knows when her hair is done it’s time to walk. They have a very beautiful relationship”* (Female, 38 years old, Grade 7).

*“So, because I’m working and he’s not currently, he does take on more responsibilities than I do. So he drops her at crèche, he fetches her at crèche, he baths her sometimes, most of the time, he ensures that her favourite movie is put on so, he makes her bottles most of the time, I’m kinda blessed in that aspect… he interacts with her, and plays a lot with her, more than I do uhm, because he knows mommy is always busy”* (Female, 25 years old, Degree).

## 4. Discussion

The parents and caregivers in this study understand what their children need to thrive within the first 1000 days of life. They highlighted the contribution of the type and quality of care and support given to children to enhance their future development. The significance of receiving parental care and support has been reported in the literature, as over 250 million children under the age of five years in low- and middle-income countries may not reach their developmental potential due to various reasons [11,12]. Poor parenting practices have been reported to be associated with negative emotional outcomes in children [44]. Additionally, parents with a poor understanding of child development are less likely to identify developmental delays in their children [45]. International organisations such as the World Bank, UNICEF and WHO, have identified parenting education programmes as a priority for improving child development outcomes in low- and middle-income countries [13].

The participants acknowledged that children need parents to be responsive and warm. This form of parenting can be referred to as an authoritative parenting style. This parenting style is consistent with parents that are nurturing, warm and sensitive to the child’s needs, while constantly considering the child’s age and maturity when forming behavioural expectations [46]. An authoritative parenting style, which includes being responsive to the child’s needs, has been regarded as the approach with the best outcome for children [47]. Furthermore, responsive caregiving is a component of the Nurturing Care Framework [14]. The Nurturing Care Framework was developed to facilitate and aid children to not only survive, but to thrive, and to transform health and human potential. This shows that many of the parents may understand what they need to do to transform the health, human potential and development of their children.

The child’s needs for holistic development were identified in this study. Children need to be nurtured so that they can develop physically, emotionally, mentally and socially at the early stage of life as well as later in life [33,48]. This view has been supported by UNICEF, WHO and the World Bank Group, which led to the development of the Nurturing Care Framework [14]. Parental love is another need of a child that was identified by the participants. The need for parental love within the first 1000 days was supported by previous studies [15,27,29,49,50]. Maternal love has been reported to be associated with maternal sensitivity, which describes a mother’s capacity to recognise and interpret her child’s behaviour and to respond appropriately in a time-sensitive manner. In addition, maternal sensitivity is associated with children’s language acquisition, cognitive development, obesity, sleep, behavioural problems, social competence, emotionality, and temperament [49].

As reported in a previous study [35,51] our study reveals that children need parents as teachers–teaching their children ethical and moral behaviour and how to regulate emotions. The opportunities for early learning and cognitive stimulation provided by the parents are components of the Nurturing Care Framework [14]. Furthermore, cognitive stimulation and emotional care have been reported in a previous study as needs of children in the first 1000 days [15]. In addition, opportunities to regulate emotions are vital, as emotions form a large part of the socio-emotional domain of child development. Moreover, emotional regulation is agreed to promote mental health and well-being in individuals [52].

Parents as providers were also highlighted by the participants, which is feeding and the provision of food. Food has been identified as a part of children’s needs in a previous study [50]. It has been reported in the literature that infant nutrition within the first 1000 days has been linked with infants’ later physical growth, muscle mass development, brain development, cognitive functioning, socio-emotional adjustment, risk-taking behaviour, earning capacity, and a multitude of health-related concerns [16,17,18,19,20,21,22]. Furthermore, growth and development occur more rapidly during pregnancy and infancy than in any other period of the life span [16]. Exclusive breastfeeding [14], which forms part of the nutritional need of a child within the first 1000 days of life, has been associated with positive developmental outcomes [53,54,55] and a lower incidence of obesity, hypertension and high cholesterol [55]. Therefore, exclusive breastfeeding is strongly advocated within this period. 

This study also highlighted the importance of a father in the life of a child from an early stage. The children need their fathers to thrive [35]. Fathers are considered to be a role model for both boys and girls, as well as being teachers and providers for the family. They are also involved in parenting, however most times as co-parents. Studies show that children who co-reside with their biological fathers have higher self-esteem [56]. In addition, studies have shown that fathers influence vital decisions after childbirth. These include whether women breastfeed, how long they breastfeed for and how timeously they register their infants’ birth [55]. Furthermore, fathers who are involved in the early days of their children’s life are more likely to remain involved throughout a child’s life. These fathers are also more likely to take on shared child care and responsibility [57]. 

### 4.1. Implications of the Study

This study provides insight into parents’ and caregivers’ understanding of what their children need to thrive within the first 1000 days in low-income communities of South Africa. We discovered that parents understand what they need to do to enhance the developmental outcomes of their children. This study will inform good practice in Early Childhood Development. The study will assist policymakers to develop appropriate policies to support parents and for them to provide what they have identified their children need to thrive. Additionally, this study will guide service providers to provide appropriate services for parents to improve the developmental outcomes for children within the first 1000 days of life. This is important, as many children in low-income countries may not reach their development potentials. Moreover, there may be internal and external factors (such as poverty, substance abuse, crime and unemployment) that may prevent the parents from providing what they have identified their children need to thrive. 

### 4.2. Recommendations

This study shows that parents and caregivers in low-income communities understand what their children need to thrive within the first 1000 days of life. Although parents know how to ensure that their children thrive, it does not necessarily mean that they, living in a low-income or resource constrained community, have the abilities and resources to do it. Therefore, there is a need to understand how to address the internal and external factors within the communities which may prevent parents from providing what children need to thrive within the first 1000 days of life. For example, how, within the first 1000 days of life, can fathers be more participative with mothers from pregnancy through birth and in child care? This understanding may facilitate the development of policies and the availability of services for parents, which include fathers. The government should develop policies and programmes that will remove or reduce internal and external factors preventing parents from performing their functions within the community, especially in resource-constrained communities. Service providers including non-profit organisations should also direct their services toward eliminating these factors.

### 4.3. Strengths and Limitations of the Study

The strength of the study is that a qualitative method was used to collect data. This allowed an in-depth exploration of the phenomenon under study. Another strength of the study is that women who had children between 0 and 2 years of age were interviewed. This enabled an understanding of the phenomenon under study, from those who were directly affected. One of the limitations of this study is that all the participants were women, except one. Another limitation is that other stakeholders such as government officials, service providers and community leaders were not interviewed. 

## 5. Conclusions

The first 1000 days is a crucial period in a child’s life and requires an integrated approach to achieve positive childhood outcomes. The Nurturing Care Framework is a guide for parents, caregivers, and stakeholders to ensure children thrive and have their lives transformed. The findings of this study show that positive parenting, specifically authoritative parenting practices will lead to positive child development outcomes and well-being. Furthermore, parents living in these low-income communities understand children’s needs for holistic development; how parents are needed to teach and guide their children in terms of moral and ethical behaviour and be providers in terms of nutrition, play, love and care. Therefore, this study shows that parents understand what children need to thrive in the first 1000 days of life. In addition, an exploration of the internal and external factors within the community which may prevent parents from providing what their children need, is recommended. The study could assist policymakers and service providers to design appropriate interventions for parents living within these communities.

## Figures and Tables

**Table 1 children-08-00483-t001:** Participant characteristics.

Characteristics of Parents and Primary Caregivers	Participants (30)
Gender	
Female	29
Male	1
Age (years)	
16–30	17
31–45	9
46–60	3
61–75	1
Highest of Level Education	
Never Attend school	2
Grade (0–7)	2
Grade (8–11)	20
Grade (12)	4
Diploma	1
Degree	1
Gender of Child	
Unknown	1
Male	13
Female	16
Age of Child (months)	
0–6	6
7–12	11
13–18	2
19–24	11

## Data Availability

The data that support the findings of this study are available within the manuscript.

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
