# Peer review of "Exploring the Perspectives of South African Parents and Primary Caregivers Living in Low-Income Communities on What Children Need to Thrive within the First 1000 Days of Life"

_children, 2021, doi:10.3390/children8060483_

Round 1

Reviewer 1 Report

Thank you for giving me the opportunity to act as a reviewer for this article.

The research problem and aim (to explore the perspectives of South African parents and primary caregivers living in low-income communities on what children need to thrive within the first 1000 days) is well justified and its relevance is adequately argued. However, the research problem should also be defined as a question.

The quotes in the findings add value to the work. As a qualitative researcher, I understand that we do not quote all the responses given by the subjects, but only those that overlap with the needed information to the greatest extent. The parents who were interviewed appear to be the “perfect parents”.

The findings may be significant to the formative practices (not only to develop appropriate policies to support parents to provide what their children need to thrive, or to provide appropriate services for parents to improve the developmental outcomes of their children within the first 1000 days of their life). I suggest including this idea as well in the implications of the study.

In lines 461-462 you say that “Also, the nature of the study, which is qualitative may affect the generalisation of the study”. In general, in qualitative studies, generalizations are not valid, so you should explain this statement.

The Materials and Methods, Results and Discussion sections are excellently written. However, you should clarify why in line 459 you say that “One of the limitations of this study is that all the participants were 459 women”, while in Table 1 there are 29 female and 1 male.

Author Response

Open Review

(x) I would not like to sign my review report

() I would like to sign my review report

English language and style

() Extensive editing of English language and style required

() Moderate English changes required

() English language and style are fine/minor spell check required

(x) I don't feel qualified to judge about the English language and style

Yes         Can be improved              Must be improved           Not applicable

Does the introduction provide sufficient background and include all relevant references?

(x)           ( )           ( )           ( )

Is the research design appropriate?

(x)           ( )           ( )           ( )

Are the methods adequately described?

(x)           ( )           ( )           ( )

Are the results clearly presented?

(x)           ( )           ( )           ( )

Are the conclusions supported by the results?

(x)           ( )           ( )           ( )

Comments and Suggestions for Authors

Thank you for giving me the opportunity to act as a reviewer for this article.

POINT 1: The research problem and aim (to explore the perspectives of South African parents and primary caregivers living in low-income communities on what children need to thrive within the first 1000 days) is well justified and its relevance is adequately argued. However, the research problem should also be defined as a question.

RESPONSE 1: Thank you for point this out.

ACTION 1: It has been defined.  Page 3, lines 110-111.

POINT 2: The quotes in the findings add value to the work. As a qualitative researcher, I understand that we do not quote all the responses given by the subjects, but only those that overlap with the needed information to the greatest extent. The parents who were interviewed appear to be the “perfect parents”.

RESPONSE 2: Thanks for saying this.

ACTION 2: We did select participants based on pre-determined criteria. Page 4, lines 146-156.

POINT 3: The findings may be significant to the formative practices (not only to develop appropriate policies to support parents to provide what their children need to thrive, or to provide appropriate services for parents to improve the developmental outcomes of their children within the first 1000 days of their life). I suggest including this idea as well in the implications of the study.

RESPONSE 3: Thanks for your suggestion.

ACTION 3: It has been added. Page 10, lines 462-463.

POINT 4: In lines 461-462 you say that “Also, the nature of the study, which is qualitative may affect the generalisation of the study”. In general, in qualitative studies, generalizations are not valid, so you should explain this statement.

RESPONSE 4: Thanks for pointing this out

ACTION 4: The sentence has been removed because we do not have any justification for the statement. In qualitative studies, generalizations are not valid.

POINT 5: The Materials and Methods, Results and Discussion sections are excellently written. However, you should clarify why in line 459 you say that “One of the limitations of this study is that all the participants were 459 women”, while in Table 1 there are 29 female and 1 male.

RESPONSE 5: Thanks for pointing this out.

ACTION 5:  It has been corrected. Page 10, line 501.

Submission Date

01 May 2021

Date of this review

17 May 2021 12:29:42

Reviewer 2 Report

The manuscript presents the results of research on the perspectives of South African parents and primary caregivers, who live in low-income communities on what children need to thrive within the first thousand days of life. The authors presents the results of qualitative research conducted with the use of interviews among parents of children up to 2 years of age.

In terms of methodology, the text was prepared well. I suggest the authors refer to the publication: Chava Frankfort-Nachmias, David Nachmias, Research Method in the Social Sciences, Worth Publisher 2008 instead of to Earl Babbie, The basic of social research, Wadsworth Cengage Learning, 2014 – which is a well-known textbook for students.

The authors clearly specified the main research problem. I would like to know what were the specific problems? I did not find information when the research was conducted.

The authors used well-known publications in their research. However, one could refer to the publications of the most famous developmental psychologists. I wonder if there are similar results from research conducted in Europe or Asia?

Numerous repetitions appear in the manuscript.

I would like ask the authors to compare the results of their research with similar studies conducted in higher income communities.

I have to emphasize that the recommendations are very good. It can be noted that these recommendations are also justified in other regions of the world, for example in creating social policy for groups of emigrants and refugees.

The text requires minor editorial corrections:

  • Abstract, p. 1: “participants participated” – I suggest “respondents participated”,
  • 2 – there is no need to write the full name of UNICEF or WHO – everyone knows that organizations,
  • 3 – “United State(s) of America,
  • 5 – Data analysis – dot at the end,
  • 6 – Theme 2. penultimate sentence – “especially mental health” – dot.
  • 7 – Moral and values citation – third line – dot before “Knowing”.

These comments do not affect the overall rating of the manuscript, which is good overall.

I recommend the manuscript for publication after taking into account the comments of the reviewer.

Author Response

Open Review

(x) I would not like to sign my review report

( ) I would like to sign my review report

English language and style

( ) Extensive editing of English language and style required

( ) Moderate English changes required

(x) English language and style are fine/minor spell check required

( ) I don't feel qualified to judge about the English language and style

Yes         Can be improved              Must be improved           Not applicable

Does the introduction provide sufficient background and include all relevant references?

( )           (x)           ( )           ( )

Is the research design appropriate?

( )           (x)           ( )           ( )

Are the methods adequately described?

(x)           ( )           ( )           ( )

Are the results clearly presented?

( )           (x)           ( )           ( )

Are the conclusions supported by the results?

( )           ( )           (x)           ( )

Comments and Suggestions for Authors

POINT 1: The manuscript presents the results of research on the perspectives of South African parents and primary caregivers, who live in low-income communities on what children need to thrive within the first thousand days of life. The authors presents the results of qualitative research conducted with the use of interviews among parents of children up to 2 years of age.

RESPONSE 1: Thank you.

ACTION 1: No action. 

POINT 2: In terms of methodology, the text was prepared well. I suggest the authors refer to the publication: Chava Frankfort-Nachmias, David Nachmias, Research Method in the Social Sciences, Worth Publisher 2008 instead of to Earl Babbie, The basic of social research, Wadsworth Cengage Learning, 2014 – which is a well-known textbook for students.

RESPONSE 1: Thank you for your suggestion.

ACTION 1: We have used the recommended text.  Page 4, line 145.

POINT 3: The authors clearly specified the main research problem. I would like to know what were the specific problems? I did not find information when the research was conducted.

RESPONSE 1: Thank you for point this out.

ACTION 1: It has been added.  Page 5, line 176.

POINT 4: The authors used well-known publications in their research. However, one could refer to the publications of the most famous developmental psychologists. I wonder if there are similar results from research conducted in Europe or Asia?

RESPONSE 1: Thank you for point this out.

ACTION 1: It has been added.  Page 3, lines 94-111.

POINT 5: Numerous repetitions appear in the manuscript.

RESPONSE 1: Thank you for point this out.

ACTION 1: It has been corrected.

POINT 6: I would like ask the authors to compare the results of their research with similar studies conducted in higher income communities.

RESPONSE 1: Thank you for point this out.

ACTION 1: It has been compared.  Page 9, lines 391-457.

POINT 7: I have to emphasize that the recommendations are very good. It can be noted that these recommendations are also justified in other regions of the world, for example in creating social policy for groups of emigrants and refugees.

RESPONSE 1: Thank you for point this out.

ACTION 1: No action taken.

The text requires minor editorial corrections:

POINT 8: Abstract, p. 1: “participants participated” – I suggest “respondents participated”,

RESPONSE 1: Thank you for point this out.

ACTION 1: It has been corrected.  Page 1, line 19.

POINT 9: 2 – there is no need to write the full name of UNICEF or WHO – everyone knows that organizations,

RESPONSE 1: Thank you for point this out.

ACTION 1: It has been corrected.  Page 2, line 58.

POINT 10: 3 – “United State(s) of America,

RESPONSE 1: Thank you for point this out.

ACTION 1: It has been corrected.  Page 3, line 97.

POINT 11: 5 – Data analysis – dot at the end,

RESPONSE 1: Thank you for point this out.

ACTION 1: It has been corrected.  Page 5, lines 189.

POINT 12: 6 – Theme 2. penultimate sentence – “especially mental health” – dot.

RESPONSE 1: Thank you for point this out.

ACTION 1: It has been corrected.  Page 7, line 250.

POINT 13: 7 – Moral and values citation – third line – dot before “Knowing”.

RESPONSE 1: Thank you for point this out.

ACTION 1: It has been defined.  Page 3, lines 322.

These comments do not affect the overall rating of the manuscript, which is good overall.

I recommend the manuscript for publication after taking into account the comments of the reviewer.

Submission Date

01 May 2021

Date of this review

13 May 2021 13:35:14